# Global and Regional Implications of Biome Evolution on the Hydrologic Cycle and Climate in the NCAR Dynamic Vegetation Model

## Jessica Levey * and Jung-Eun Lee

Department of Earth, Environmental, and Planetary Sciences, Brown University, Providence, RI 02912, USA; leeje@brown.edu
* Correspondence: jessica_levey@alumni.brown.edu; Tel.: +1-508-330-6522

**Abstract:** Vegetation influences climate by altering water and energy budgets. With intensifying threats from anthropogenic activities, both terrestrial biomes and climate are expected to change, and the need to understand land–atmosphere interactions will become increasingly crucial. We ran a climate model coupled with a Dynamic Global Vegetation Model (DGVM) to investigate the establishment of terrestrial biomes starting from a bareground scenario and how these biomes influence the climate throughout their evolution. Vegetation reaches quasi-equilibrium after ~350 years, and the vegetation establishment results in global increases in temperature (~2.5 °C), precipitation (~5.5%) and evapotranspiration as well as declines in albedo and sea ice volumes. In high latitude regions, vegetation establishment decreases albedo, causing an increase in global temperatures as well as a northward shift of the Intertropical Convergence Zone (ITCZ). Low latitude tropical afforestation results in greater evapotranspiration and precipitation, and an initial decrease in temperatures due to evaporative cooling.

**Keywords:** DGVM; biome; evapotranspiration; precipitation; albedo; ITCZ; low-latitude; high-latitude

## 1. Introduction

The vegetation–climate nexus is a key component for understanding land–atmosphere interactions. Vegetation influences climate by its function in biogeophysical cycles, including the transfer of water vapor, energy, and trace gases from land to atmosphere, as well as modifying physical properties such as albedo and surface roughness [1]. Albedo varies with land surface types because darker, vegetated land absorbs more solar radiation than snow-covered or bareground land [2]. Transpiration from plants influences the partitioning of the turbulent surface energy budget [3] and provides moisture to the atmosphere. Water vapor is an important greenhouse gas, and studies show that increased transpiration from more vegetation may increase atmospheric water vapor content, and in turn global temperature [4]. Local increases in evapotranspiration rates may decrease local temperature because of evaporative cooling [5]. Many studies have shown the importance of transpiration in determining precipitation [6]. Vegetation uses carbon dioxide during photosynthesis, and biomes may act as carbon reservoirs. Uptake of atmospheric $CO_2$ concentration by vegetation plays a part in global temperature and climate [7].

Vegetation plays a role in large-scale climate dynamics. The global atmospheric energy balance and sea surface temperature (SST) gradients influence the position of the Intertropical Convergence Zone (ITCZ) [8]. With hemispheric temperature asymmetry, the ITCZ migrates towards the warmer hemisphere [9], and the position of the ITCZ varies with changes in cross-equatorial atmospheric

heat transport [10]. Biome location and distribution plays a key role in global climate by determining albedo, which influences hemispheric energy budgets prompting the movement of the ITCZ [11].

The cumulative effects of small-scale plant processes are key determinants in biome establishment, an essential component of the climate–vegetation nexus. Individual plants differ in influence on the carbon cycle due to varying rates of stomatal conductance, the exchange of $CO_2$ intake for water loss during transpiration, as well as respiration and photosynthesis rates [6]. Ecophysiological properties such as leaf, stem and root dimensions, growth efficiency and water storage capacity, influence climate through their varying abilities to generate exchanges between terrestrial biomes and the atmosphere. Processes vary based on plant type. For example, trees influence climate very differently to grasses because of their deeper roots, faster rates of photosynthesis and respiration, and larger water storage capacity [12]. Grasses typically establish in drier climates because the shallow roots allow them to outcompete other plant types for limited rainwater availability. In wetter climates, trees establish because their deeper roots allow them to retrieve water from deeper soil layers over longer periods of time [12].

Plant functional types, as well as vegetation distribution over varying latitudinal gradients influences climate responses. For example, shrub establishment in the tundra causes the albedo to decrease, leading to increases in net radiation and subsequently causing further snowmelt [2]. Boreal forests establishment in high-latitude areas has a large effect on global climate due to the significance of the 'snow/ice/albedo' feedback systems [13]. Additionally, the conversion of biome types will influence surface temperature, albedo, roughness, and precipitation [14]. Low-latitude land-use changes influence climate, particularly precipitation patterns, and alter the partitioning of heat from turbulent surface fluxes of sensible to latent heat [15]. Biomes are important for other geophysical processes and play a role in the carbon cycle and the storage capacity of atmospheric carbon in soil and vegetation [16].

Biome evolution and establishment influence global climate due to their cumulative effect of the chemical and physical processes from individual plants. Historically, biomes have been defined by temperature, precipitation, and evapotranspiration [17]. Plants influence land–atmosphere interactions through photosynthesis, which drives the facilitation and partitioning of gases, nutrients and water vapor between land and atmosphere. Surface roughness and albedo also alter Earth's geophysical processes by their impact on the boundary condition and absorption of incoming radiation. The crucial role of vegetation in the land–atmosphere nexus influences the global conservation of energy, momentum, and water vapor [1]. Biomes [18] influence the global energy and water cycles, influencing large-scale circulation and precipitation patterns [11].

Land Surface Models (LSMs) provide the fluxes of energy, water vapor and momentum to the atmosphere, and vegetation properties are often fixed using monthly vegetation cover derived from the satellite data [19]. However, to be able to predict future climate conditions, it is necessary to develop a model that includes the change in vegetation dynamics. The Dynamic Global Vegetation Model (DGVM) quantifies interactions between individual plants as well as plant communities on a grid cell level [20], including mortality, disturbance, competition, decomposition, recruitment, allocation, and establishment [20]. DGVMs allow the simulation of shifting biomes as a response to climate conditions and vegetation growth changes [18], and the model can also calculate the rate of physical and biological changes of plant functional types (PFTs) [21]. The DGVM does not yet include several essential variables that dictate PFT establishment and migration, such as seed dispersal mechanisms and urban barriers for plant migration [18].

Other studies have focused on the climate–vegetation relationship; however, we were able to utilize climate modelling and create a simulation run to experimentally test and quantify the influence of biome evolution on global and regional climates. To better understand the vegetation–climate relationship, it is essential to investigate how local and large-scale climate dynamics respond to biome evolution. Here we investigate how biome evolution plays a role in global and regional climate responses. We ran a climate model coupled with a DGVM, starting from a bareground initial condition,

to investigate (a) how biomes evolve and establish throughout time and latitudes; (b) how biome establishment influences temperature, precipitation, albedo and evapotranspiration on local and global scales. To look at regional changes, we chose three highly vegetated regions varying in vegetation type and latitude to study: northeastern South America just south of the equator, west Africa just north of the equator, and midwestern North America. Regional climate patterns may exemplify changes in global energy transport and atmospheric circulation, as well as shifts in the ITCZ.

## 2. Materials and Methods

We used the National Center for Atmospheric Research Community Earth System Model (NCAR CESM 1.2.2) [22] to analyze how the vegetation establishment quantitatively influences global climate. We ran CESM 1.2.2 consisting of a slab ocean, atmosphere, and land model [13]. The slab ocean configuration allows full interaction between atmosphere and ocean mixed layer but does not allow any ocean circulation change. The thermodynamic response of sea ice is accounted for in our model setup.

Land surface model is coupled to the Dynamic Global Vegetation Model (DGVM), incorporating dynamic vegetation responses to physical and biogeochemical changes, including the carbon and nitrogen cycles, in the physical climate system. The model simulation started from a bareground initial condition and ran for 400 years, until each biome type reaches a quasi-equilibrium state. We investigated changes in global and regional climate patterns as a response to biome evolution and increased vegetation coverage. NCAR DGVM includes 14 PFTs, and we aggregated the PFTs into three different categories of biomes: trees, shrubs and grass. More details about the DGVM simulations, including the comparison with the observations, can be found Levis et al. (2004) [20].

The model was run using pre-industrial carbon dioxide concentrations of ~280 ppm. As a result, there is no feedback coming from the changing carbon cycle. The model's horizontal resolution is longitude of 2.5° and latitude of 1.9°. The atmosphere model has 26 atmospheric layers, and the land model has 10 soil layers.

## 3. Results and Discussions

### 3.1. Global Biome Evolution

The first plant functional type to establish was grass, next shrubs, and last was trees (Figure 1). Grass was the first PFT to emerge and increased rapidly, within the first 5 years, due to lack of competition and remains the predominant global biome type for the first 20 years of the simulation. We note that the area of grass tends to be underestimated with this model [20]. Once shrub and tree coverage increases, grass area declines as a result of competition. Around year 10 tree and shrub coverage begins to increase. Around year 15, grasses begin to steeply decline because it is outcompeted by shrubs and trees (Figure 1a). The area of shrubs increases at a fast rate until year 50, when it hits quasi-equilibrium. Shrubs mainly establish in subtropical and mid-latitude regions. Around the year 50, the rate of tree coverage increases, but the forested area is still growing until it reaches quasi-equilibrium around year 350.

Global-scale biome establishment largely impacts climate. Increased vegetation contributes to the feedback loops among vegetation increase, albedo decrease, warming temperatures and an increase in water vapor [14]. Trees are the dominant biome at the quasi-equilibrium state, evolving from low latitude regions up to higher latitude regions. Starting at year 2, tropical tree coverage increases the fastest and earliest, then temperate trees begin to evolve shortly after at a slower rate. Boreal trees are the last to establish, around year 10, and take the longest to reach equilibrium because they generally are in higher latitude areas and they grow slowly (Figure 1b). The tropical trees developed very rapidly for around 25 years and reached quasi-equilibrium by about year 40, whereas the temperate and boreal trees had slower rates of establishment and took more time to reach equilibrium. Temperate trees reached quasi-equilibrium after about 100 years, whereas boreal trees began to approach equilibrium

around 350 years. Similar patterns occurred with different tree phenology types; the broadleaf trees established more quickly and reached an equilibrium faster than the needleleaf trees.

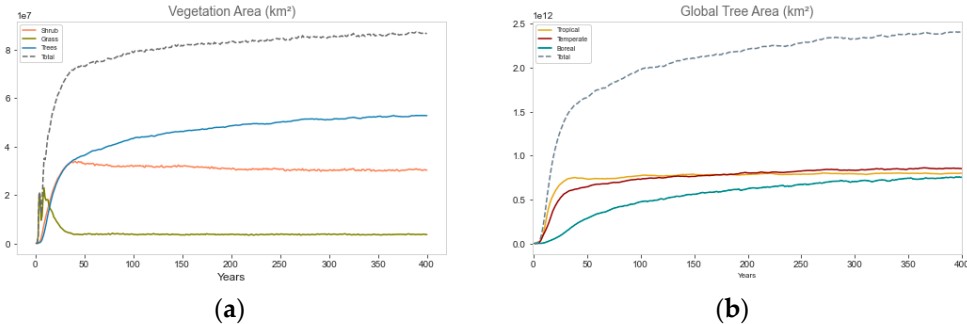

(a)　　　　　　　　　　　　　　　　　　　　　　　　(b)

**Figure 1.** (**a**) The vegetation area of each category of plant functional type: shrub (orange), grass (green), and trees (blue) and total (grey dashed) over 400 years. (**b**) Dissection of tree evolution area based on tree type, including tropical (yellow), temperate (maroon), boreal (teal), and total tree area (grey dashed).

Initially, forests established in low-latitude regions, and spread closer into mid-latitude regions and up into the northern hemisphere. By the end of the 400-year run, trees established on most available landmass, excluding deserts and polar areas (Figure 2c). Grasses developed in mostly mid and low latitude regions and decreased over the rest of the run once areas of shrubs and trees began to increase. Shrubs evolved in regions of relatively low precipitation, in the southern and eastern parts of northeastern South America, and in west Africa. Throughout the 400-year period, shrubs distributed over all latitudes and remained in regions adjacent to forests. Based on the vegetation area maps, we chose three regions, with highly vegetated land coverage of each varying plant functional type (Figure 2). At year 400, there is more vegetated area in the northern hemisphere, which is a result of more landmass availability for plants to evolve into.

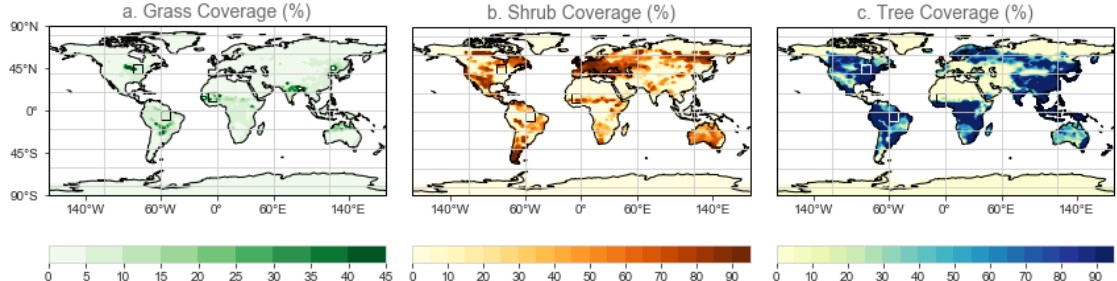

**Figure 2.** Global vegetation coverage after 400 years of the model running from a bareground scenario. The maps of percent land coverage of plant functional type categories: (**a**) grass; (**b**) shrubs; and (**c**) trees. The boxes are the three regions that were analyzed: midwestern North America, northeastern A Runs were produced by running the Dynamic Global Vegetation Model (DGVM) starting from a bareground simulation. Maps produced by coupling a slab-ocean model and present-day land-ocean configurations, atmospheric composition, and solar energy budget with the National Center for Atmospheric Research Community Earth Systems Model (NCAR CESM) 1.2.2 (Neale et al., 2013).

*3.2. Global Climate Response*

Rapid global biome establishment and evolution occurred in the first 35 years of the simulation, after starting from a bareground scenario (Figure 1). As a result, temperature and precipitation responded to the Earth's surface becoming vegetated. The initial condition of the model explains the initial increase in sea ice, prior to the model's-sea ice distribution decreasing throughout the rest of the simulation run (Figure 3). Land surface became darker than bareground in areas where vegetation established, and in turn the land absorbs more solar radiation (Figure 3b). The decreased albedo from

biome establishment induced sea ice melt (Figure 3c), further perpetuating temperature increases (Figure 4a) due to the increase in vegetated area as well as the decrease in sea ice area (Figure 3c).

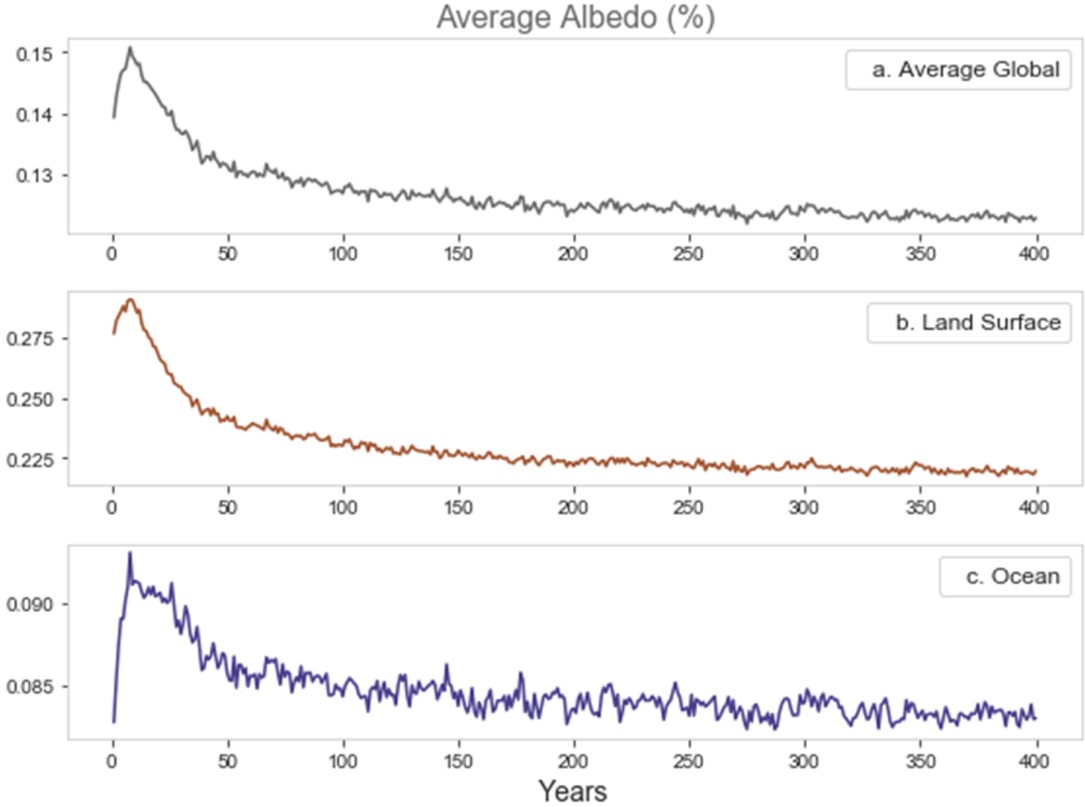

**Figure 3.** Surface albedo changes over the first 400 years broken down into: (**a**) Average global albedo; (**b**) land surface albedo; (**c**) ocean albedo.

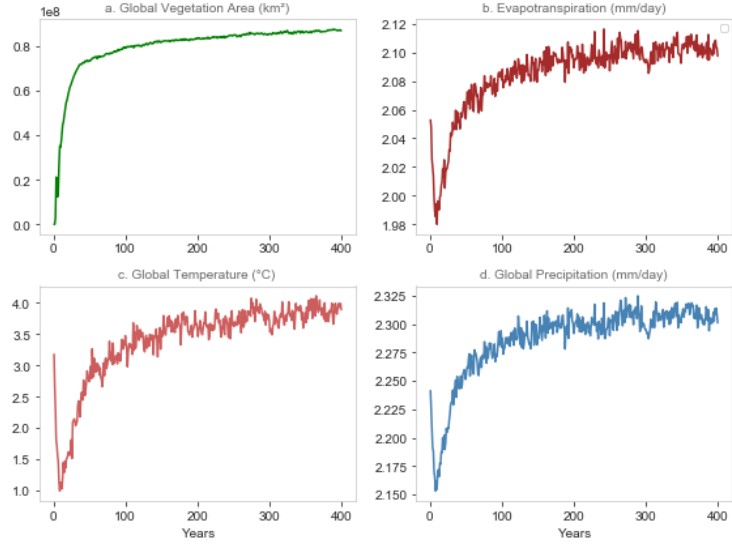

**Figure 4.** Average global climate responses after running the 400 years of the model running from a bareground scenario. (**a**) global vegetated area (km$^2$) (**b**) evapotranspiration (mm/day); (**c**) global temperature (°C); and (**d**) precipitation (mm/day).

The global response to biome establishment prompts an increase in precipitation, temperature, and evapotranspiration (Figure 4), and a decrease in albedo and sea ice coverage (Figure 3). In the

first 10 years, global trends oscillate due to the initial condition of the model. After this initial phase, the climate trends correspond with the presence of vegetation. Counting from the coldest point when vegetation has not yet been established, global temperature increases ~2.5 °C because of biome establishment. Evapotranspiration recycles moisture and induces precipitation, so these values begin to reach equilibrium at the same time as vegetated area does, around year 10. Global precipitation increase is ~5% (2.725 to 2.875 mm/day). Furthermore, increased temperature and precipitation increase rates of photosynthesis, which drives plant growth. Global temperature, precipitation and evapotranspiration rates approach an equilibrium state after approximately 100 years; the same time as the tree area begins to level off (Figure 1a).

This timing suggests that trees may be the dominant biome type that dictates global climate. Most of the climate parameters peak during the times when trees, specifically broadleaf and temperate and/or tropical tree areas increase (Figure 1b). Vegetation coverage rapidly increases in the first 50 years, which is when the surface temperature spikes and albedo decreases (Figure 4). As a feedback to decreasing albedo and increasing temperature, sea ice volume also decreases during this 50-year period (Figure 3). Between years 50 and 400, temperature, precipitation, and evapotranspiration trends continue to increase but the rate of change slows down, corresponding to the rate of change in trees (Figure 1b).

Afforestation and biome establishment in high-latitudes alters biogeophysical fluxes, increasing water vapor from plant evapotranspiration as a result of the greenhouse effect [23]. Temperature increases from high-latitude afforestation is also a result of the positive feedback loop between decreased albedo and increased snowmelt. Low-latitude temperature change is relatively small because increases in energy and local cooling due to the increase in evapotranspiration cancel out.

Globally, vegetated land coverage increases at the steepest rate from years 0–50 but continues to increase over the next 50 years but the rate of change slows down (Figure 1a). Between years 50 and 100, grass and shrub biomes approach equilibrium and tree biome development rate slows down. In Figure 5, the differences of the means were calculated between the two intervals (years 50–100 and 0–50) for leaf area index (LAI), surface temperature, precipitation and albedo. The regions where leaf area index change is positive (Figure 5a), albedo is mostly negative (Figure 5d). Change in surface temperature is positive in both hemispheres and over most latitudes (Figure 5b). During this 50 year interval, the ITCZ moves northward—over time, precipitation in the Southern Hemisphere decreases and precipitation in the Northern Hemisphere increases. Equatorial precipitation belts show increases in precipitation in the tropical regions north of the equator and decreases in precipitation in the equatorial regions south of the equator (Figure 5c). A greater percentage of the Northern hemisphere becomes vegetated, causing a precipitation differential from the asymmetrical global energy transport.

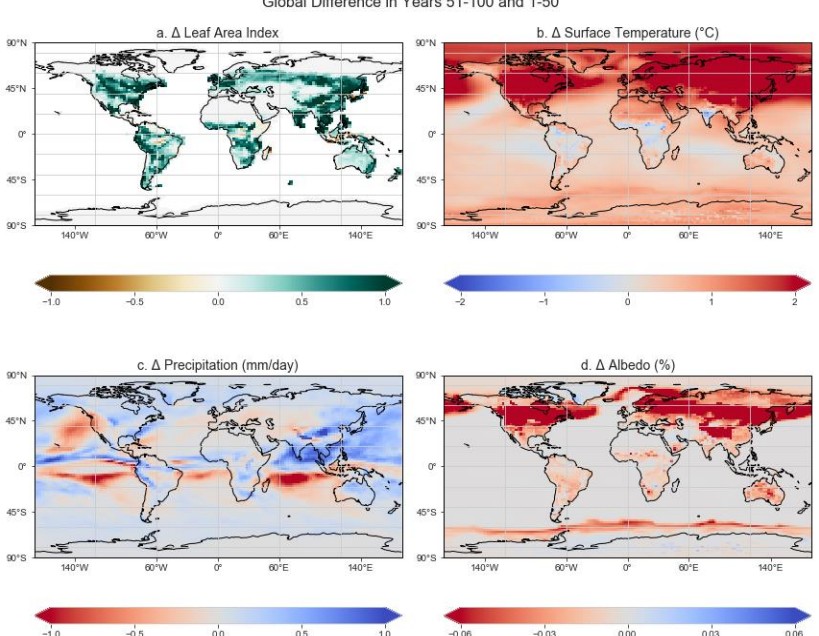

**Figure 5.** Global maps of changes in 50-year intervals between years 1–50 and 51–100. Global maps of: (**a**) Δ Leaf Area Index (LAI), (**b**) Δ surface temperature (°C), (**c**) Δ precipitation (mm/day) and (**d**) Δ albedo (%). Maps produced by coupling a slab-ocean model and present-day land-ocean configurations, atmospheric composition, and solar energy budget with the National Center for Atmospheric Research Community Earth Systems Model (NCAR CESM) 1.2.2 (Neale et al., 2013).

### 3.3. Latitudinal Responses

Over time, the difference in mean precipitation over the first 100 years (Years 50–100 minus 0–50) has the largest values due to the fast increase in vegetation (Figure 6a). Low- and mid-latitude regions north of the equator experience more precipitation and increased peak precipitation. The subtropical areas in the Southern Hemisphere undergo decreasing precipitation averages throughout the 350 years. Over time, precipitation and albedo values stabilize with biomes reaching equilibrium. More plants are in the Northern Hemisphere, which means that the albedo decrease and temperature increase is greater in the Northern Hemisphere (Figure 6). As biomes evolve, the ITCZ shifts more towards the northern hemisphere generating a shift in the global water cycle and energy balance. Afforestation in mid and high latitudes initiated a shift in the ITCZ as well as an increase in temperatures in the Northern Hemisphere due to increased absorption of solar radiation and a greater capacity for soil water storage and availability [17]. The shift in ITCZ can influence regional climate in the tropics [8]. With greater solar radiation absorption the surface energy budget increases, driving sensible and latent heat fluxes [11]. Increased temperature and water vapor prompts increased recycling of water vapor from the land to the atmosphere leading to increased precipitation [11].

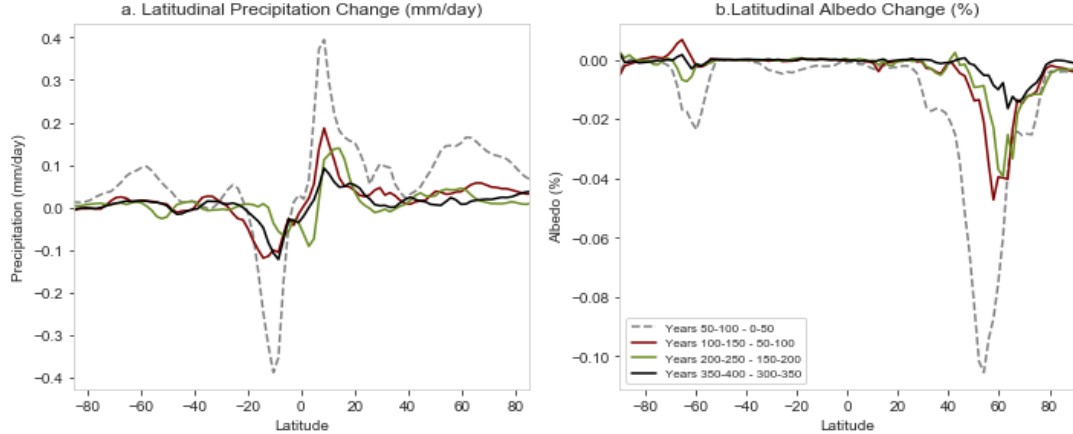

**Figure 6.** Integrals of 50-year intervals between: (**a**) precipitation values (mm/day) and (**b**) surface albedo (%) verse latitudes. Intervals between years 51–100 minus 0–50, 100–150 minus 50–100, 200–250 minus 150–200, and 350–400 minus 300–250.

### 3.4. Regional Climate Response

To better understand climate–vegetation relationships for different biomes and locations, we investigated regional climate responses to the establishment of biomes. Over the same 400-year period, we studied three regions: northeastern South America (tropical rainforests), west Africa (dry forests and shrubs), and midwestern North America (temperature forests and grasses) (Table 1).

**Table 1.** Regional Biome composition.

|  | Midwestern North America | Northeastern South America | West Africa |
|---|---|---|---|
| Latitude Range | −100° to −85° | 0° to −15° | 7° to 13° |
| Longitude Range | 45° to 55° | −60° to −50° | 2° to −12° |
| Vegetation | Temperate forest | Tropical Rainforest | Dry forest |
|  | Grasses | Trees | Shrubs |

Based on vegetation types at quasi-equilibrium.

The duration of biome establishment was determined by the rate of photosynthesis, and thus regional proximity to the equator. The northeastern South American region, vegetated predominantly by tropical forests, was the first of the three regions to become vegetated, beginning at year 2 and increasing until quasi-equilibrium was met around year 40 (Figure 7). Around year 8, plants began to show up in the midwestern North American and west African regions. However, the west African region had far less vegetated coverage of about 17 percent, probably due to the low precipitation averaging to ~0.14 mm/day (Figure 7a). Plant growth is rapid in the west African region at the beginning of the run, but does not reach equilibrium until around year 140. Grasses and shrubs were initially established in the west African region, after about 15 years trees were established in the region. The potential for biome establishment and vegetated land coverage corresponds more closely to precipitation gradients than it does to temperature in the low latitude regions. Precipitation increased in this region as a result of the ITCZ migration from more vegetation cover in the Northern Hemisphere, and shrubs start to slowly increase until year 140. Although competition takes place between plant function types, tree establishment is inhibited by precipitation. At the end of the simulation, trees were the dominant vegetation type in northeastern South America, and grasses were the dominant vegetation type in midwestern North America. The midwestern North American region, furthest from the equator, had the most vegetated coverage (Figure 7a) once equilibrium was reached around year 60.

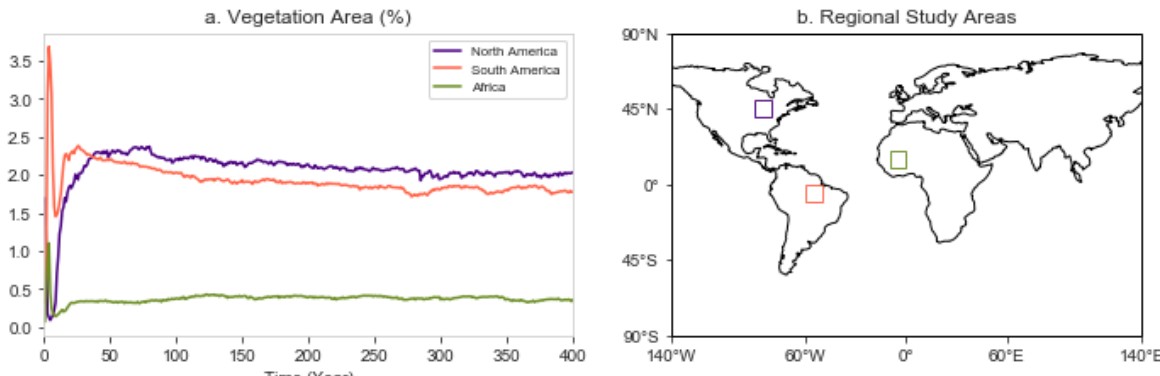

**Figure 7.** (**a**) Percentage of global vegetation coverage by regions: West Africa (green), northeastern South America (red), and midwestern North America (blue). (**b**) The three regional areas mapped.

### 3.4.1. Northeastern South America

The study region of northeastern South America, particularly the northeastern part of Brazil, is influenced by the shifting of the Intertropical Convergence Zone (ITCZ) [8]. The vegetation in this region consists mostly of tropical trees because grass and shrubs were outcompeted by trees. As the vegetation coverage in this area rapidly increases in the first 25 years (Figure 7), the albedo decreases because the darker color of the forest reflects less solar radiation compared with bareground. Although this region exemplifies similar vegetation coverage and albedo trends to the global trends, the regional temperature and precipitation patterns are not consistent with the global trends (Figure 8). Regional temperature decreases with increased tree presence due to evaporative cooling and more convection, which occurs in forested regions. Although global precipitation increases, due to the shift of the ITCZ northwards, precipitation decreases in the southern hemisphere equatorial regions. With less precipitation, vegetation growth may be inhibited, and the total vegetation coverage reaches a quasi-equilibrium around this time. In the northeastern South America region, local evapotranspiration and cloud cover play a critical role in regulating temperature.

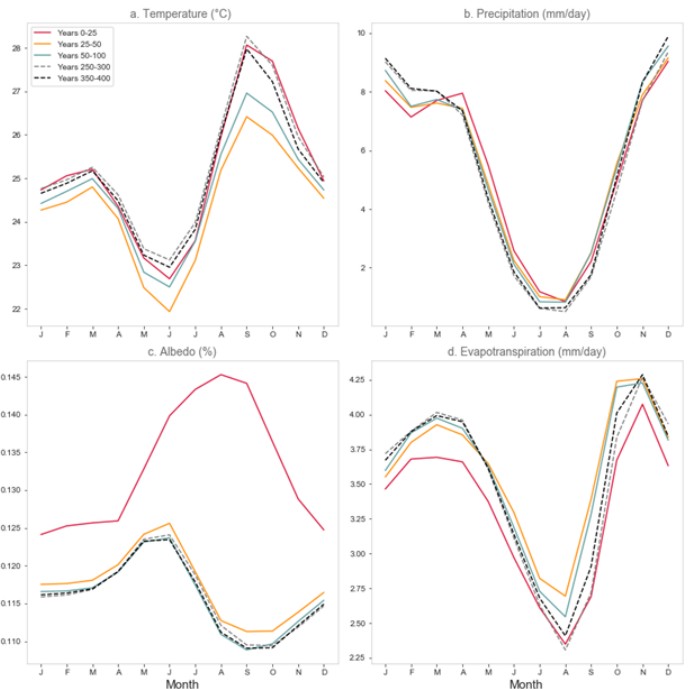

**Figure 8.** The annual cycle average from regional northeastern South America for: (**a**) surface temperature (**b**) precipitation (**c**) albedo and (**d**) evapotranspiration. Averages of climate response data were taken between years 0–25 (red), before PFTs began to establish, and 25–50 (gold), during the largest increases in vegetated area. Averages were also taken from years 50–100 (green), 250–300 (gray dashed) and 350–400 (black dashed).

Vegetation establishment prompts an increase and a slight shift in the seasonality of evapotranspiration and precipitation in northeastern South America. Global temperature spiked in the first 50 years of the model run (Figure 4a); however, in northeastern South America, temperature decreased between the first and second 25-year interval. Due to tree establishment, rates of evapotranspiration increase (Figure 8) and regional temperatures decrease due to evaporative cooling [24] (Figure 8a). Globally, precipitation increases, but this region has precipitation decrease in the dry season and increase in the wet season (Figure 8b). Resulting from the Northern hemisphere becoming largely vegetated, precipitation increased in boreal winter and decreased during boreal summer resulting from northward movement of the ITCZ and shifts in the Hadley Circulation. The ITCZ shift was particularly exemplified in the northeastern South American region due to its location just south of the equator. Over the 400 years, precipitation in this region decreased during the boreal summer and increased during the boreal winter (Figure 8b). During the boreal winter, mean temperature and precipitation values slightly increased and mean temperature slightly increased throughout the 400-year run, probably due to the increase in global temperature.

### 3.4.2. West Africa

The second region that was studied is west Africa, located north of the equator. The regional climate responses differ from the northeastern South American regional climate trends due to hemispheric energy balances and transport. Between years 0–25 and 26–50, the vegetated area increased most, which corresponds to the largest increases in precipitation and albedo (Figure 9). As vegetation coverage increases, evapotranspiration increases throughout all seasons. Moreover, temperature initially increases between February and June, but decreases between July and December (Figure 9a) when precipitation and evapotranspiration increases (Figure 9b,d). Precipitation increases, particularly in the peak season, indicate the northward shift of the ITCZ (Figure 6).

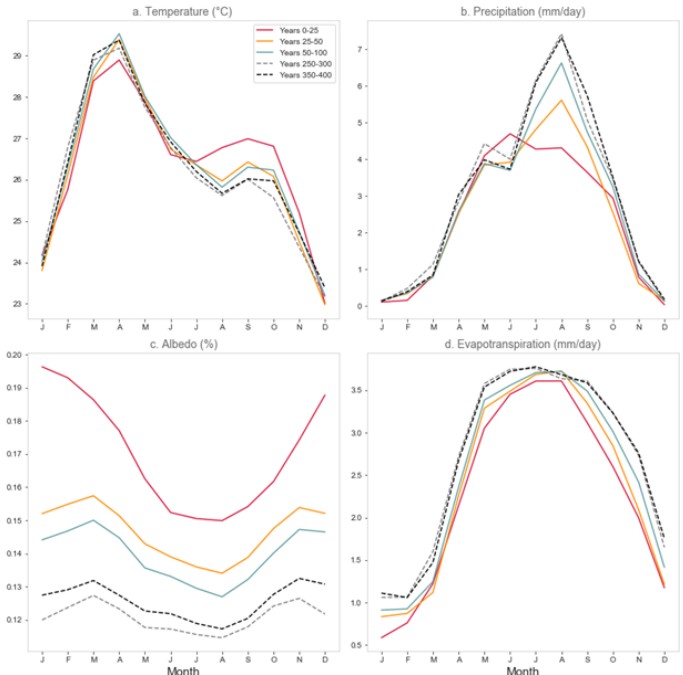

**Figure 9.** The annual cycle average from regional west Africa for: (**a**) surface temperature (**b**) precipitation (**c**) surface albedo and (**d**) evapotranspiration. Averages of climate response data were taken between years 0–25 (red), before PFTs began to establish, and 25–50 (gold), during the largest increases in vegetated area. Averages were also taken from years 50–100 (green), 250–300 (gray dashed) and 350–400 (black dashed).

In the wet season of July, August and September, the mean temperature does not change much throughout the 400 years, but the mean precipitation increases by about 2 mm/day (more than 30%) over this period. In the dry season of December, February and March, the mean temperature and precipitation values do not change much.

### 3.4.3. Midwestern North America

In the midwestern North American region albedo decreases as the system shifts bareground to grass and from grass to tree domination (Figure 7). Much like the global-scale relationship, as albedo decreases due to vegetation establishment, temperature increases throughout the 400-year period. Additionally, evapotranspiration and precipitation increase clearly during this time as well. Temperature increases in this region and in turn both evapotranspiration and precipitation rates increase (Figure 10). Throughout the 400-year period, the albedo effect prompts the greatest increase in temperature to occur during the winter (Figure 10a). Precipitation and evapotranspiration rates are highest in the summer months, the optimal time for plant growth; thus, enhancing precipitation (or water availability) during optimal growing time further promotes vegetation growth in this area, and allows for a positive feedback reaction.

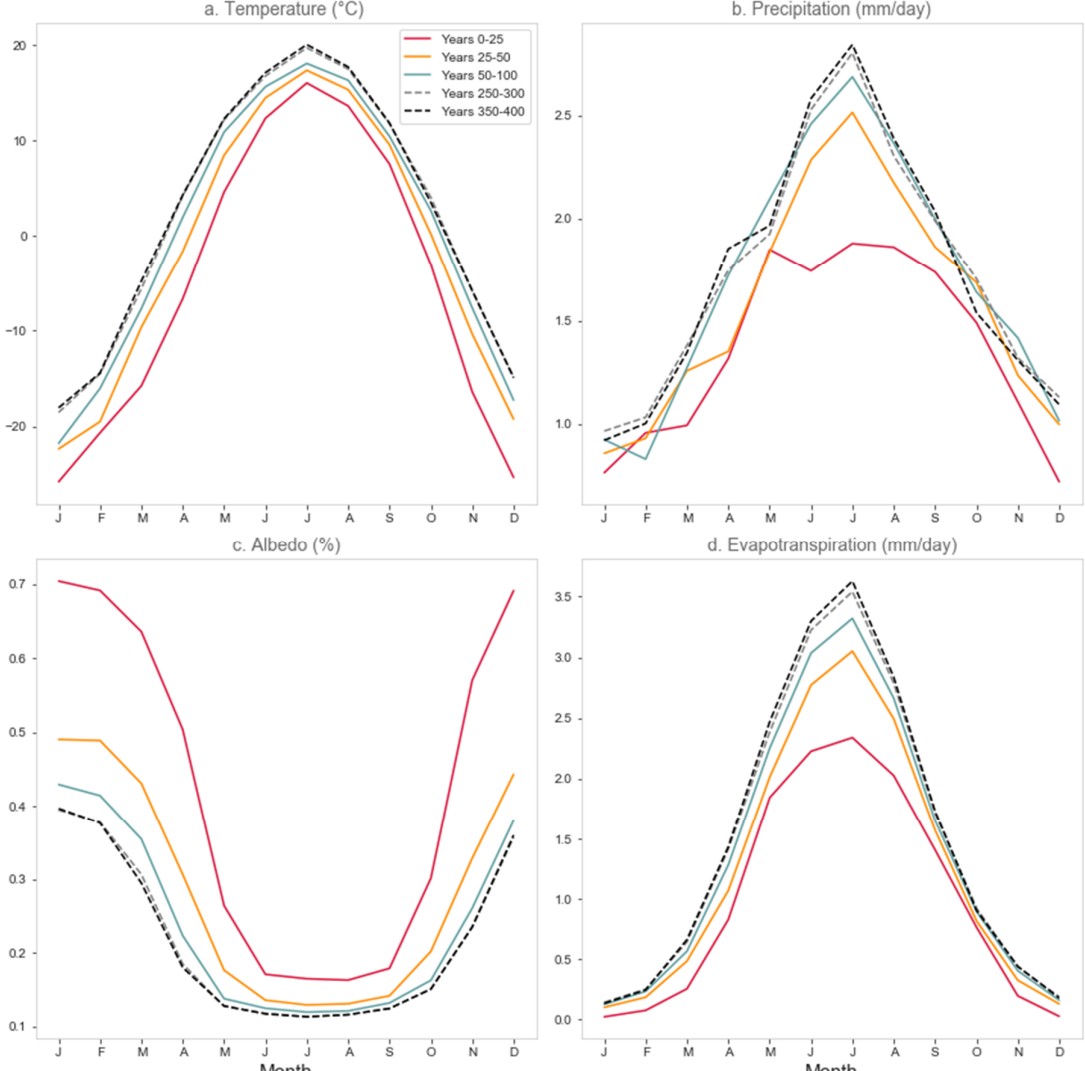

**Figure 10.** The annual cycle average from midwestern North America for: (**a**) surface temperature (**b**) precipitation (**c**) albedo and (**d**) evapotranspiration. Averages of climate response data were taken between years 0–25 (red), before PFTs began to establish, and 25–50 (gold), during the largest increases in vegetated area. Averages were also taken from years 50–100 (green), 250–300 (gray dashed) and 350–400 (black dashed).

In this mid-latitude region, afforestation prompts an increase in temperature due to decreased albedo and greater absorption of incoming radiation, as well as from the greenhouse effect from greater concentrations of atmospheric water vapor (Figure 10a,b) [25,26]. Linear increases in temperatures appeared in the regions with vegetation establishment, whereas the temperature increase fluctuates more in the northeastern South America region, where local evapotranspiration and cloud cover play a critical role in regulating temperature.

To demonstrate the different seasonal response to the vegetation establishment in midwestern North America, we plotted the evolution of the seasonal temperature and precipitation (Figure 11). Around year 100 when total vegetation reached a potential quasi-equilibrium (Figure 3) the precipitation values in the summer begin to increase (Figure 11b). The midwestern North American region corresponds to the global trends of vegetation establishment prompting increased temperature and precipitation. In this region, temperature and precipitation increased far more during the winter months than it did in the summer months (Figure 11). Wetter and hotter summer months promote

further plant growth in this region and perpetuate the feedback between increased vegetation and wetter and hotter regions.

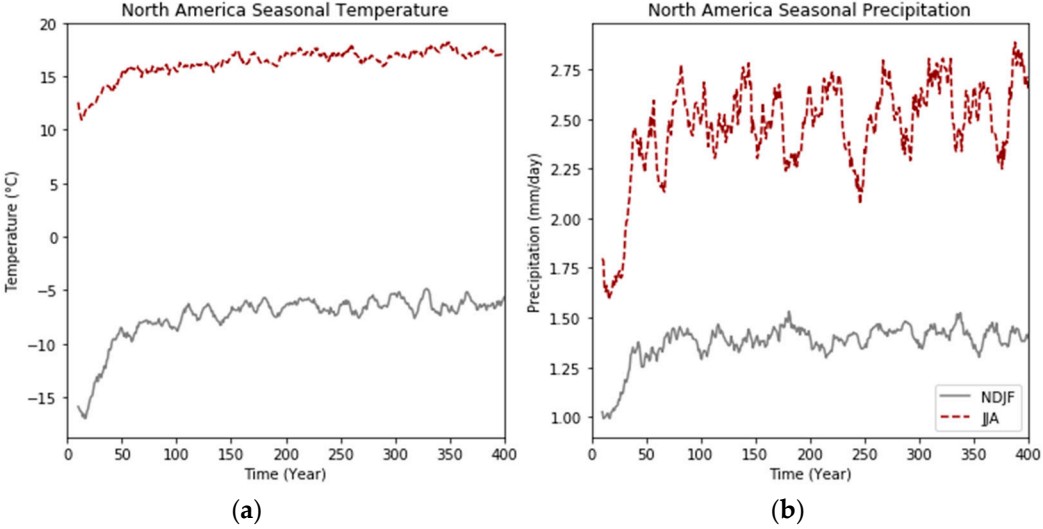

**Figure 11.** The ten-year moving averages from the peak warm months, JJA, (red) and from the cold season, NDJF, (grey) of midwestern North America of (**a**) midwestern North America seasonal temperature and (**b**) midwestern North America seasonal precipitation over 400 years. The temperature and precipitation data were averaged over the wet season months (June–August) and the dry season (November-February) months. Coordinates of the region of midwestern North America that was studied are latitudes 15° to 55° and longitudes of −130° to −70°.

## 4. Conclusions

Land surfaces are changing over time due to plant evolution on geological time scales and anthropogenic forces such as farming, deforestation, and urbanization in recent millennia. Climate change can influence the land surface on many time scales. Vegetation influences climate through several biogeophysical and biogeochemical mechanisms. To understand how vegetation establishment and evolution impacts climate on a global and regional scale, we ran a climate model coupled with a DGVM to examine the responses of temperature and precipitation to vegetation changes. Global biome establishment and evolution prompts global temperature (~2.5 °C), precipitation (~5%), and evapotranspiration to increase, and albedo and sea ice area to decrease. Trees are the primary driver of global climate variables and start at low latitude regions and evolve into higher latitude regions. Due to more land surface availability in the Northern Hemisphere, vegetation evolves primarily into the north. As a result, the effect of albedo decreases and the net energy gain is higher in the Northern Hemisphere, and the ITCZ shifts northwards. The northeastern South American region exemplified that low latitude regions are influenced the most by changes in evapotranspiration. High latitude regions are mostly influenced by changes in albedo, particularly in winter, a consequence of the trees covering snow effect, as shown in the midwestern North American region.

Regional responses sometimes contradicted global trends based on PFT and proximity to the equator. The northeastern South American and west African regions were located on separate hemispheres and experienced opposite trends in precipitation. The midwestern North American region showed that as vegetation evolved into the region, the conditions became more optimal for plant growth, further promoting vegetation development. Global albedo responds to the increase in trees and the changes in land surface cover. Although climate models allow experimental investigations of relationships, model outputs have accuracy limitations [27] at finer resolutions. I suggest further examination of vegetation as a driver of climate processes at a smaller scale, with improved accuracy and precision compared to current model capabilities.

**Author Contributions:** Investigation, J.L. and J.-E.L.; Validation, J.-E.L.; Writing—original draft, J.L.; Writing—review & editing, J.-E.L. All authors have read and agreed to the published version of the manuscript.

**Funding:** This research was funded by NSF AGS-1944545.

**Acknowledgments:** We thank H. Kershaw at Brown University's Center for Computation and Visualization for help with running the model. We thank Brown University's Computer and Information Services for help with computer equipment. We thank M. Wu, W. Xu and E. Mischell for help reviewing earlier versions of this paper.

**Conflicts of Interest:** The authors declare no conflict of interest.

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
