# Peer review of "Global and Regional Implications of Biome Evolution on the Hydrologic Cycle and Climate in the NCAR Dynamic Vegetation Model"

_land, doi:10.3390/land9100342_

Round 1

Reviewer 1 Report

This study examined how global and regional climate systems respond to the evolution of vegetation by investigating the vegetation and climate interactions. The climate model coupled with DGVM were simulated for biome evolutions starting from a bareground initial condition to a quasi-equilibrium state.

A manuscript is well-written with a scientific merit to improve the knowledge in the land and atmosphere interactions. The publication of manuscript is recommended after revising it as in the following specific comments.

Emphasize the merit of this study: Before mentioning the research objectives of this study (Line 87-96 of manuscript), the merit of this study, compared to the previous studies, should be emphasized. I think this is the place to stress how this study is different from the previous works. I could see the merit of this study to examine the effects of ‘vegetation evolution’ on climate.

L 93: The reasons for selecting three study regions need to be mentioned. It has described in L 163-164, but it’s better to mention earlier.

L 112: While the related paper was mentioned, how the model outputs are consistent with observations should be described at least for the climate variables (e.g. albedo, temperature, precipitation, ET) used in this study.

L 116: ‘The model was run for a 400-year period…a quasi-equilibrium state’ can be removed, because it was mentioned in L 107.

L 149: Remove the comma after slowly.

Study region boxes: To provide the vegetation information of three study regions along with geographical information, it could be shown in Figure 2a for North America, Figure 2b for Africa, and Figure 2c for South America.

L 197: Global temperature increase of 4.5 degree C needs to be checked. The temperature change from 12 to 14.5 degree C with ~2.5 degree C increase is shown in Figure 4a. The number in L 439 is also to be revised if needed.

Figure 4d: LAI has a peak around year 25 and then declines by year 100. After year 100 it gradually increases by year 400. The evolution of LAI is different from that of climate variables and also from vegetation and tree areas as in Figure 1. Why the evolution of LAI is different from other variables? It’s useful to mention about this in the manuscript.

L 230: The evolution of biome in high-latitudes should be associated not only with greenhouse effect, but with the positive feedback of snow and albedo. The feedback needs to be mentioned here.

Figure 5: It’s suggested to change the order of figures in Figure 5 to follow the biogeophysical processes, i.e. (a) LAI, (b) albedo, (c) surface temperature and (d) precipitation.

L 300-302: Where the number of 17% was from? In Figure 7a the max is 3.5%. It’s needed to be checked if Figure 7a is vegetation area or precipitation. Accordingly, Figure 7a should be revised.

L 302-312: Which figure supports the results mentioned here? The related figures should be represented in the manuscript.

Figures 8, 9, 10: The figures of annual evolutions of climate variables, averaged over each study region, from year 0 to 400 are needed to support the descriptions in manuscript. The figures could be included here or as in supplementary document if the space is limited.

Figures 8: In the title of figure, ‘annual’ should be changed ‘monthly’ or ‘annual cycle’.

L 370-373: Suggest to modify this to explain the reason of the previous sentence as including the effects of ITCZ northward movement on precipitation change in the South America region.

L 391: Remove ?? after 30.

L 395: Change 3.7 to 3.4.3.

L 424: Change season to seasonal

L 425: It happened around year 100 not year 50. Figure 11: Indicate (a) and (b) in figures.

L 426: Change Figure 11 to Figure 11b

L 428: ‘far more’ Can you quantify this? The slope value of linear regression can be used to quantify it.

L 454: I strongly suggest adding the concluding remark including any limitation of study, suggestion of further study etc.

Author Response

Emphasize the merit of this study: Before mentioning the research objectives of this study (Line 87-96 of manuscript), the merit of this study, compared to the previous studies, should be emphasized. I think this is the place to stress how this study is different from the previous works. I could see the merit of this study to examine the effects of ‘vegetation evolution’ on climate.

We elaborated on how our work compares to other studies (line 86-88) I emphasized that our experiment and how we are experimentally testing the vegetation-climate relationship by running a global climate model using a DGVM coupled with NCAR’s CESM 1.2.2 to quantify this relationship.

L 93: The reasons for selecting three study regions need to be mentioned. It has described in L 163-164, but it’s better to mention earlier.

We added an explanation in the Introduction (line 94-95) describing our reasoning for selecting these regions.

L 112: While the related paper was mentioned, how the model outputs are consistent with observations should be described at least for the climate variables (e.g. albedo, temperature, precipitation, ET) used in this study.

Study region boxes: To provide the vegetation information of three study regions along with geographical information, it could be shown in Figure 2a for North America, Figure 2b for Africa, and Figure 2c for South America.

We added the region outlines to Figure 2 in order to show geographically where the regions are and also to show the vegetation composition of each of the places that we selected for our study.

L 197: Global temperature increase of 4.5 degree C needs to be checked. The temperature change from 12 to 14.5 degree C with ~2.5 degree C increase is shown in Figure 4a. The number in L 439 is also to be revised if needed.

2.5 degrees C increase is the correct values.  The values were revised throughout the paper to match this value that was in the abstract.

Figure 4d: LAI has a peak around year 25 and then declines by year 100. After year 100 it gradually increases by year 400. The evolution of LAI is different from that of climate variables and also from vegetation and tree areas as in Figure 1. Why the evolution of LAI is different from other variables? It’s useful to mention about this in the manuscript.

Instead of using LAI for Figure 4, I used the total vegetated coverage plot because it is a much clearer comparison for the point we are trying to make.  LAI has a bit more uncertainty that is not necessary for the central relationships that we are trying to describe.

L 230: The evolution of biome in high-latitudes should be associated not only with greenhouse effect, but with the positive feedback of snow and albedo. The feedback needs to be mentioned here.

This point was added in line 211.

Figure 5: It’s suggested to change the order of figures in Figure 5 to follow the biogeophysical processes, i.e. (a) LAI, (b) albedo, (c) surface temperature and (d) precipitation.

Figure 5 was changed to follow this order and to have vegetated coverage replace the LAI plot.

L 300-302: Where the number of 17% was from? In Figure 7a the max is 3.5%. It’s needed to be checked if Figure 7a is vegetation area or precipitation. Accordingly, Figure 7a should be revised.

The percentage of vegetated coverage was quantified earlier in the study, but since the figure was not a central part of the study, we did not include it.  We took out these percentages and clarified that the vegetated area for this figure is calculating the fraction of global vegetated coverage that is occurring within these regional extents.

L 370-373: Suggest to modify this to explain the reason of the previous sentence as including the effects of ITCZ northward movement on precipitation change in the South America region.

We clarified the point that the relationship between the global shift in the ITCZ was corresponding to the changes that were happening in the South American region (Lines 347-349)

L 454: I strongly suggest adding the concluding remark including any limitation of study, suggestion of further study etc.

We added a statement about the limitations and future hopes for this area in line 438-439.

Reviewer 2 Report

Review of “Global and Regional Implications of Biome Evolution on the Hydrologic Cycle and Climate in the NCAR Dynamic Vegetation Model” by Jessica Levey and Jung-Eun Lee, submitted to Land sections: Land–Climate Interactions, Land Use and Climate Change (932523)

This paper focuses on an experiment with a global climate model regarding land surface - atmosphere (LA) interaction knowledge during the first 400 years when vegetation develops around the world and in three specific selected regions. The paper is essential for future group research using this model, or eventually, independent research focused on model development and the LA interaction community. Like how well written and concise the paper is, I encourage authors to share this relevant global data, increase visibility, and collaborate.

This work is suitable for the Journal Land in sections Land–Climate Interactions or Land Use and Climate Change, after revision. I detailed specific major and minor comments in red and orange in the pdf, respectively, while typos/corrections/recommendations are in yellow. Apologize for English mistakes and any comment that could appear as rude.

Here, some general questions whose answers should be added to the manuscript to guide the reader.  

  • Do other researches perform similar experiments with other GCMs, or is this the first one?
  • Who are the main readers?
  • Which are the next steps?
  • What could be the main applications of the study?

Below, some general comments that summarize many comments I did in the pdf.

  • It is unclear what the message is in paragraphs 1, 3, 4, and 5 in the Introduction.
  • In my opinion, the selection of the South American region is not adequate.
  • There are many typos/inconsistencies are referring to figures, regions, and results in general.

Author Response

Do other researchers perform similar experiments with other GCMs, or is this the first one?

We added a sentence to the introduction (line 86-88) explaining that this idea has of vegetation influences climate has been studied, but it has not been experimentally tested by looking at biome evolution and establishment

Who are the main readers? What could be the main applications of the study? Which are the next steps?

We added this information to parts of the introduction and conclusion to talk about how we are expanding off of typical climate modeling simulations. The study is targeted towards other climate modelers in the field or other natural scientists who study land-atmosphere interactions in order to emphasize the importance of vegetation when considering climate and climate change.   In the conclusion we added our thoughts on next steps (line 438-439), which would include improving accuracy and precision of model output at a smaller scale in order to further test this relationship.

In my opinion, the selection of the South American region is not adequate.

We tried to clarify the importance of northeastern South America by relating the observations we saw to the global observations of the ITCZ shift (Lines 347-349).  Additionally, we added a statement (line 20) about the temperature trends that were occurring in the northeastern South American region were different than the global trends.

There are many typos/inconsistencies are referring to figures, regions, and results in general.

We implemented your comments on the pdf about inconsistencies with regions by referring to the regions as ‘west Africa’, ‘midwestern North America’ and ‘northeastern South America’ throughout the paper to clarify that we are talking about these specific regions of study rather than the whole continent. Additionally, we cleaned up figures 4, and 8-10 in order to clarify some of the inconsistencies that were pointed out.
